# Zero-Padding and Spatial Augmentation-Based Gas Sensor Node Optimization Approach in Resource-Constrained 6G-IoT Paradigm

**DOI:** 10.3390/s22083039

**Published:** 2022-04-15

**Authors:** Shiv Nath Chaudhri, Navin Singh Rajput, Saeed Hamood Alsamhi, Alexey V. Shvetsov, Faris A. Almalki

**Affiliations:** 1Department of Electronics Engineering, Indian Institute of Technology (BHU), Varanasi 221005, Uttar Pradesh, India; shivnathchaudhri.rs.ece17@iitbhu.ac.in; 2Software Research Institute, Technological University of the Shannon, Midlands Midwest, N37HD68 Athlone, Ireland; salsamhi@ait.ie; 3Faculty of Engineering, IBB University, Ibb 70270, Yemen; 4Department of Operation of Road Transport and Car Service, North-Eastern Federal University, 677000 Yakutsk, Russia; alekssvetlanamax@gmail.com; 5Department of Transport and Technological Processes, Vladivostok State University of Economics and Service, 690014 Vladivostok, Russia; 6Department of Computer Engineering, College of Computers and Information Technology, Taif University, Taif 21944, Saudi Arabia; m.faris@tu.edu.sa

**Keywords:** electronic nose, gas sensor array, sixth-generation wireless communication technology (6G), 6G-IoT, zero-padding, spatial augmentation, convolutional neural networks, machine learning, artificial intelligence, pattern recognition

## Abstract

Ultra-low-power is a key performance indicator in 6G-IoT ecosystems. Sensor nodes in this eco-system are also capable of running light-weight artificial intelligence (AI) models. In this work, we have achieved high performance in a gas sensor system using Convolutional Neural Network (CNN) with a smaller number of gas sensor elements. We have identified redundant gas sensor elements in a gas sensor array and removed them to reduce the power consumption without significant deviation in the node’s performance. The inevitable variation in the performance due to removing redundant sensor elements has been compensated using specialized data pre-processing (zero-padded virtual sensors and spatial augmentation) and CNN. The experiment is demonstrated to classify and quantify the four hazardous gases, viz., acetone, carbon tetrachloride, ethyl methyl ketone, and xylene. The performance of the unoptimized gas sensor array has been taken as a “baseline” to compare the performance of the optimized gas sensor array. Our proposed approach reduces the power consumption from 10 Watts to 5 Watts; classification performance sustained to 100 percent while quantification performance compensated up to a mean squared error (MSE) of 1.12 × 10^−2^. Thus, our power-efficient optimization paves the way to “computation on edge”, even in the resource-constrained 6G-IoT paradigm.

## 1. Introduction

Advances in low-power single-board processors and controllers have ushered a revolution in the growth of Internet of Things (IoT) based industrial applications [1]. Low power sensor nodes with edge intelligence are essential in resource-constrained sixth-generation wireless communication technology (6G)-driven IoT scenarios [2,3]. Gas sensor systems play a vital role in various IoT ecosystems in different areas, viz., environmental monitoring [4], healthcare [5], agriculture [6] and foods and beverages [7,8]. Non-selective gas sensor-based gas sensor systems carry significant redundancy within their signature responses. Removal of sensor elements based on their redundancy can be an approach to reduce the actual power consumption of the sensor node [9]. Optimizing the sensor node’s effectively consumed power using such an approach will make 6G-IoT-based applications more acceptable for industrial automation. Designing and optimizing intelligent sensor nodes on edge is stringently desired in 6G-driven sensor networks. These optimization techniques are applied to sensor nodes at hardware and software levels [9,10].

The 6G-IoT has specifically considered the sensor nodes to be ultra-low power. It is one of the significant key performance indicators in the 6G-IoT paradigm. Tremendous 6G-IoT device and network station deployment results in massive energy consumption and increasing CO emissions. The goal of green 6G-IoT is to design and create energy-efficient sensor nodes, communication strategies, and protocols through optimization [3,11,12]. For example, in Industry 4.0, optimizing energy consumption, QoS (throughput, latency, capacity, and so on), and resource use in 6G smart applications and automation systems [13]. Green 6G-IoT systems for smart environments would benefit from energy conservation strategies, renewable energy supplies, and energy harvesting methods.

Somov et al. [9] have optimized the performance of a gas sensing node using a circuit designed for the accurate measurement of gas concentration. They used two gas sensing elements on a Wheatstone bridge in the sensing circuit. The authors later optimized the circuit using a single gas sensing element on a voltage divider. Furthermore, Amarlingam et al. [10] used compressed sensing for data aggregation at the data processing level to achieve energy efficiency in resource constrained IoT systems. Thus, the researchers have worked on sensor node optimization at various levels, viz., at the level of circuit designing utilizing multiple configurations and at the data processing level using compression technique to data aggregation.

IoT systems designed to detect/measure hazardous gases are found more effective in ubiquitous real-time monitoring of the indoor and outdoor ambiances for people’s safety [14]. Real-time monitoring of the presence of toxic and hazardous gases, especially in industrial ambiance, is of serious concern for the safety and welfare of the workers [15,16,17,18]. The use of non-selective metal oxide semiconductor (MOX)-based gas sensor elements typically make the sensor node cheaper yet power-hungry. Reducing the number of physical gas sensors used in forming the gas sensor array can surely help in reducing the gas sensor node’s power requirements. However, such reduction is possible only when any such reduction in the number of sensor elements does not produce a significant deviation in the performance of the sensor node itself. With the introduction of Artificial Intelligence (AI)-based “computation on the edge” and data pre-processing-based analysis spaces, certain power saving can be expected [19]. Convolutional Neural Networks (CNNs) have been found very effective for such edge computing [20].

Recently, Peng et al. [21] proposed a two-dimensional Deep Convolutional Neural Network (DCNN) with 38 layers called GasNet for gas classification. The architecture of GasNet is relatively deeper for gas classification, exerting high computational costs. Zhao et al. [22] utilized a one-dimensional DCNN to classify specific binary mixtures of ethylene, CO, and methane. Wei et al. [23] have improvised LeNet-5 [24] to classify CO, Methane, and their binary mixture using a computation savvy approach. However, aforesaid implementations are relatively complex and unsuitable in resource-constrained situations requiring low power sensor nods with edge computation and detecting a wide range of gases. While classifying the gases using an extensively drifted dataset, a hybrid CNN architecture outperforms by curtailing drift effects [25].

This work considers a gas sensor node consisting of four gas sensing elements for power-efficient optimization suitable for the 6G-IoT resource-constrained scenarios. We have optimally reduced the effective power consumption of the considered sensor node by removing the most redundant gas sensor element(s). Furthermore, we have mitigated any performance degradation after the removal of certain sensor elements by using intelligence on the edge and advanced data pre-processing approaches. CNNs have been used to develop high-performance intelligent gas sensor systems due to their inherent feature extraction capability [21,22,23]. We have successfully optimized the gas sensor node to have only two physical gas sensing elements and maintained the performance by using zero padding, spatial augmentation, and CNN. The removed sensor element is zero-padded to create a 2D shape of the input vector to CNN. This primary 2D input was then upscaled into a 6 × 6 input vector, using spatial augmentation. A simpler CNN is then used to receive the upscaled 6 × 6 input to classify and quantify the unknown test samples of the considered gases with very high accuracy.

The proposed hypothesis for gas sensor node power optimization has been tested under two scenarios viz., (i) three physical gas sensors and one zero-padded virtual sensor element and (ii) two physical gas sensors and two zero-padded virtual sensor elements, while the performance of the four-element gas sensor system has been taken as the baseline for the comparison.


**Scenario 1: Three physical and one zero-padded virtual sensor element**


In this scenario, one out of the four physical gas sensors were replaced by a zero-padded value. The 2 × 2 sensor array signatures, so generated, were then spatially augmented into a 6 × 6 input data vector. These zero-padded spatially augmented input vectors were processed using a simpler CNN for the classification and quantification of the 16 unknown test samples of the considered gases. Successful removal of one out of four gas sensor elements provides 25 percent power-saving without any significant deviation of the sensor node’s performance. In this scenario, we can have four combinations for the sensor element removal.


**Scenario 2: Two physical and two zero-padded virtual sensor elements**


In this scenario, two out of the four physical gas sensors were replaced by zero-padded values. By following the signature processing method as described in the previous scenario, the successful removal of two out of four gas sensor elements provides 50 percent power-saving without significant deviation. We can have six such combinations for the sensor element removal.

### 1.1. Motivation and Contributions

Low-cost high-performance detection and estimation of environmental pollution present in the ambient air in the industrial and household sector have become inevitable to deliver better health and safety of workers and residents. While 6G-IoT will support ubiquitous monitoring and data gathering from such sensor systems, it also requires sensor nodes to be low-power, low bandwidth, and high performance. The use of “intelligence on the edge” provides significant performance enhancement at a lower bandwidth. However, the reduction in consumed power of a gas sensor node has been a challenge due to design constraints posed by the MOX-based sensor elements of a gas sensor array. By leveraging the information redundancy present in non-selective gas sensor array-based signatures, CNNs can provide a further reduction in the power consumed by a gas sensor node without significant deviation in the performance. In recent literature, authors have mostly used DCNNs that are complex and difficult to be used on the edge.

Considering the aforesaid opportunities and limitations, we have proposed a novel approach, wherein a simpler CNN along with specialized data pre-processing approaches have been envisaged. The resulting gas sensor systems are intelligent, low-power, and low bandwidth, typically suitable for use in resource-constrained 6G-IoT applications.

To the best of the authors’ information, such an approach in gas sensor system design to deliver high accuracy in classification and quantification of a variety of gases with significant power reduction has been proposed for the first time.

The contribution of our proposed approach can be highlighted as follows:A novel approach for gas sensor node power optimization without significant compromise in performance while classifying and/or quantifying the gases.The power-efficient sensor node is a key enabler for “Computation on the edge” for resource-constrained 6G-IoT applications.The especially used data pre-processing (zero-padded virtual sensors and spatial augmentation) and CNN-based sensor node optimization can reduce the cost of hardware and effective power consumption without significant deviation in the sensor node’s performance.The proposed simpler CNN when used with the zero-padded virtual sensors and spatial augmentation is computationally less complex and is well-suited for “computations on edge” in resource-constrained 6G-IoT environments.

### 1.2. Paper Structure

The proposed work has been presented broadly under five sections. Section 1 deals with the introduction as above, while the materials and methods have been described in Section 2. Furthermore, Section 3 is devoted to the results, followed by Section 4 for discussions. Subsequently, in Section 5 conclusions are drawn.

## 2. Materials and Methods

### 2.1. The Dataset

The dataset used to demonstrate the proposed approach comprises the sensor response signatures of a four-element MOX-based gas sensor array for four gases viz. acetone (ace), carbon tetrachloride (car), ethyl methyl ketone (emk), and xylene (xyl). The corresponding array of integrated gas sensors was fabricated using thick-film technology [26]. It has four sensing elements fabricated using doping of different materials, viz., cadmium sulfide (CdS), molybdenum oxide (MoO), tin oxide (SnO_2_), and zinc oxide (ZnO). Since SnO_2_ is used as the base material, the third sensing element is an undoped sensor element. Further details on data selection and extraction can be found in [27]. They have used this data to classify respective gases using Artificial Neural Networks (ANNs).

In [28,29], a similar dataset has been utilized for the demonstration of certain data pre-processing techniques. This raw dataset consists of 58 samples (ace: 10, car: 13, emk: 18, xyl: 17) of the steady-state responses of the aforesaid gas sensor array as a whole that were used as 42 training (ace: 8, car: 10, emk: 12, xyl: 12) and 16 test (ace: 2, car: 3, emk: 6, xyl: 5) samples after pre-processing as proposed in our work. Each sample consists of a four-element vector and is labeled for its class of gas (ace, car, emk, xyl) and respective percent concentration (%Conc.). A schematic block diagram of the gas sensor array and corresponding blocks for data pre-processing and classification/quantification is shown in Figure 1a,b. Furthermore, the raw sensor characteristics of the considered gases corresponding to training samples are shown in Figure 2a–d.

### 2.2. Gas Sensor System in Resource-Constrained 6G-IoT Scenarios

Resource optimization at the sensor node level is essential in the resource-constrained 6G-IoT scenarios [30]. It is desirable that a gas sensor node should deliver high performance and consume less energy, less computational power, and utilizes optimal hardware. In resource-constrained scenarios, low power consumption enhances the viability of sensor-based mobile and wearable technologies. The evolution of 6G-IoT stringently leverages the use of edge intelligence, especially in resource-constrained environments.

The low-cost, power-efficient gas sensor systems play a vital role in various industrial and indoor applications. While using optimal resources in the resource-constrained environment, gas sensor systems are designed to provide high performance. AI-based techniques are also found suitable to further enhance the performance of a gas sensor system. Although, they also must be computationally efficient, having simpler architecture and configuration of the AI components.

Our proposed approach attempts to accomplish the stringent requirements of resource-constrained gas sensor systems. Eventually, the sensor node becomes low-cost and less power savvy. At the same time, the use of a simpler CNN makes the edge intelligent using low compute power. Non-selective gas sensors-based gas sensor systems carry significant redundancy within their signature responses. Removal of sensor elements based on their redundancy can be an approach to reduce the actual power consumption of the sensor node.

### 2.3. A 2D Convolutional Neural Network (2D-CNN)

The general architecture of a 2D-CNN is shown in Figure 3a. Generically, it has convolutional layers and pooling layers followed simply by a multilayer perceptron architecture. A 2D-CNN has been frequently used to classify the image data, where pooling layers are essentially used. Recently, various authors have started to use 2D-CNN for gas classification. However, they have used complex CNN architectures for the mentioned purpose.

Reducing the architecture complexity, we have customized a general architecture of 2D-CNN suitable for gas classification and quantification as shown in Figure 3b. It consumes less computational power and provides high-performance in classifying and quantifying the considered hazardous gases, even being simpler. Furthermore, it is applied to preprocessed data vectors obtained from raw data vectors. In this customization, pooling layers have been eliminated, which are inevitably required, while classifying the images as various pixels are mutually correlated. This customization leads to the development of a simpler CNN. The proposed simpler 2D-CNN is then trained using the corresponding training dataset. In the convolution process, we have input size 6 × 6 which convolved with the kernels of size 3 × 3 using step size 1. Thus, we achieved convolved features of size 4 × 4 at the output of the 1st convolution layer and 2 × 2 at the output of the 2nd convolution layer. These output features are flattened to a 1D vector and used as the input to the subsequent fully connected layer with 32 neurons. The resultants of this fully connected layer pass to the SoftMax activation layer for classification and linear activation layer for quantification of the considered gases. At the training stage, the stochastic gradient descent (SGD) optimizer with a learning rate of 0.001 has been used along with the hyperbolic tangent as the activation function.

### 2.4. Contextual Outline of Virtual Gas Sensors and Zero-Padding

Virtual Gas Sensors (VGSs) are the gas sensors elements in a gas sensor array, whose responses are derived from the responses of actual physical gas sensors [28]. Virtual gas sensor responses are usually derived using a variety of mathematical formulations [28]. Recently, a virtual sensor approach using principal components has been effectively used to classify the gases [31]. The inclusion of VGS adds more information for processing the gas sensor data and enhances the classification and quantification performance. In a gas sensor array, each physical element consumes a significant amount of power. MOX-based physical gas sensor elements usually respond to various gases with significant information redundancy. Therefore, removal of the physical element is possible, if we can further enhance the information by way of inclusion of virtual sensors. Accordingly, we have utilized the concept of VGS to optimize a physical gas sensor node, especially as needed in resource-constrained gas sensor systems. It is also efficient for sensor nodes working at the edge for optimal power requirements. In our approach, instead of calculating the VGS response using physical sensors, we have used zero padding and obtained good performance from the CNNs. Such incorporation does not add additional computational load. Thus, the replacement of physical sensors with zero-padded values makes the sensor array responses 2D compatible with the CNN. Reduced power and less complex CNN make the proposed approach compatible with the requirements of 6G-IoT systems operated in a resource-constrained environment.

Zero-padding is used to insert the zero values wherever it is applied. The application of zero padding is an objective-oriented task. Sometimes, it is used to pad zeros within data, and sometimes it is used to pad zeros around the data. For example, it keeps the size of the output feature equal to the input feature while using linear convolution operation [32]. Furthermore, it is also used for data vector size equalization whenever variable size data vector occurs or has missing values. The sparsing of data can also be carried out while replacing the nominal values with zeros. A comprehensive study of types of padding can be found in [33].

### 2.5. Contextual Outline of Spatial Augmentation

The spatial augmentation of data vectors enables the inputs to be compatible with 2D-CNN. The dataset used in our experiment has been obtained from a four-element gas sensor array and can be presented in 2 × 2 form of 2D response. A 2D kernel cannot operate on a 2D sensor response, therefore, by following the spatial augmentation algorithm, the 2 × 2 sensor response data is up scaled to 6 × 6 data, on which a 3 × 3 kernel can be operated very efficiently. Spatial augmentation can be applied to 2D data which can be represented in a square form, e.g., 2 × 2, 3 × 3, 4 × 4, and so on. It is shown neatly in Figure 4. Further details on spatial augmentation using mirror mosaicking can be found in [29,34]. The generalized algorithm for spatial augmentation implementation is presented in Algorithm 1.

### 2.6. Case-Based Experiments (Baseline and Possible Scenarios)

In our proposed work, we have considered a four-element MOX gas sensor array. Each MOX sensor consumes 500 mA at 5V (2.5 Watts) making the effective power requirement of 10 Watts, towards the gas sensor element only. Our proposed work reduces the actual number of physical sensors from the sensor array by using zero-padding and spatial augmentation and a simpler CNN has been trained for classification and quantification of the considered gases, paving the way to use a simpler 2D-CNN at the edge, enabling the lightweight computational intelligent, suitable for resource-constrained 6G-IoT environments.

Initially, we have four sensor elements in the gas sensor array, and the corresponding responses have been used to classify and quantify the considered gases. We have first obtained the best performing gas sensor system for accurate classification and quantification with our approach. This performance, while utilizing the responses of all four physical sensor elements, has been referred to as base performance, and the corresponding experiment is referred to as the **Baseline experiment**. Furthermore, we have replaced one physical sensor element with a zero-padded value, and then the same 2D-CNN has been used to classify and quantify the responses. This setting has been referred to as the **Scenario 1 experiment**. There are four possible cases while replacing a single physical gas sensor with zero padding. The best performing gas sensor system for accurate classification and quantification has been obtained, along with identifying the most redundant sensor element. Subsequently, we have replaced two physical sensor elements with zero-padded values. This setting has been referred to as the **Scenario 2 experiment**. There are six possible combinations while replacing two physical gas sensors with zero padding. Then, the best performing gas sensor system for accurate classification and quantification has been obtained along with recognizing the two most redundant sensor elements. Further details on the experiments are discussed in the results and discussions.
**Algorithm 1** Spatial Augmentation Procedure for Data Up-Scaling
**1****function** *Spatial Augmentation (S)*
**Input**: Number of sensor elements *S* in the sensing unit OR Length of raw data vectors
**Output**: The spatially augmented data vectors compatible with the input to the 2D-CNN**2****BEGIN****3****IF** *S* is a perfect square number:**4**Represent the data vector in the squared array of size:[√(*S*) × √(*S*)]**5**Following stage 2 as discussed, data vector spatially augmented to the size:[(3 × √(*S*)) × (3 × √(*S*))]**6****ELSE****7**Find the number of required virtual sensors to make the total number of sensor elements equal to the nearest perfect square number, say α **8**Following stage 1 virtual sensors were included to make the total number of sensor elements in the sensing unit, [(*S* + α)] **9**Represent the obtained data vectors in the squared array of size:[√(S + α) × √(S + α)]**10**Following stage 2 as discussed, data vector spatially augmented to the size[(3×√(S + α)) × (3×√(S + α))]**11****STOP**

## 3. Results

As described in the previous section, we have three experimental scenarios depending on the use of the number of zero-padded sensor elements. We designed multiple gas sensor systems under each of the experiments and compared their relative performance quotient.

### 3.1. Experiment 1: Baseline (Four Physical Sensor Elements)

In this experiment, four physical gas sensor elements have been considered in the gas sensor array. We have then captured steady-state sensor array responses for samples of the considered gases. Each sensor array response is a four-element data vector, which is then transformed into a 2 × 2 2D data vector. Subsequently, the 2 × 2 data is up scaled following the spatial augmentation algorithm, as presented in Section 2.5. This results in a 6×6 spatially augmented data vector suitable to 2D-CNN. It can be observed that the raw four-element data vectors have been scaled-up to 36-element data vectors, as shown in Figure 4. A CNN as shown in Figure 3b is then trained using the training dataset consisting of 42 × 36 input and target pairs. The trained CNN is then tested on the augmented test samples comprising of 16 × 36 input vectors, generating percentage probability of the respective class of gas sample and estimated concentration. In this baseline experiment, the class of each test sample was predicted accurately while the estimated concentration of each test sample carried a very low mean squared error (MSE). Classification and quantification performance obtained in the baseline experiment is shown in Table 1.

The best performing CNN trained and tested during the baseline experiment; we achieve 100 percent accuracy for classification. The quantification performance for the 16 unknown test samples achieves a significantly low MSE of 5.86 × 10^−3^. The sample-wise quantification performances, in this case, have been shown in Figure 5. While quantifying the gases, the maximum and minimum errors have been obtained 3.61 × 10^−2^ and 4.01 × 10^−5^, respectively. The power consumed by the gas sensor elements is 10 Watts.

### 3.2. Experiment 2: Scenario 1 (Three Physical and One Zero-Padded Sensor Element)

In this experiment, one of the four physical gas sensor elements was removed, and the respective sensor response was zero-padded (i.e., replaced by a zero value of output). Here, there are four possible cases of replacing one of the physical gas sensor elements. Furthermore, since zero-padding has been used as a replacement, it therefore does not affect the length of raw data vectors. Thus, we have a similar data vector in shape and size as obtained in the previous baseline experiment. Using the procedure shown in Figure 4, the resulting data vectors after zero padding further transformed into 6 × 6 spatially augmented data vectors generating a training dataset of size 42 × 36 and a test dataset of size 16 × 36. Thus, obtained data vectors have been used to train the 2D-CNN, as shown in Figure 3b. Similar to the baseline experiment, 42 data vectors have been used to train the CNN, while 16 unknown data vectors have been used for testing purposes. In this experiment, classification accuracy was achieved with all unknown test samples correctly classified. Furthermore, the MSEs for all possible four cases in this scenario are shown in Table 1.

We achieve 100 percent classification accuracy, not only in the best performing case, but also in all the possible cases for this scenario. Hence, with three physical and one zero-padded virtual sensor element, we have achieved 100% accurate classification. For quantification, the MSE occurs 5.82 × 10^−3^ in the best performing case, while the maximum and minimum errors achieved 3.25 × 10^−2^ and 2.93 × 10^−5^, as shown in Figure 5. With one sensor element removal, we have reduced the power consumption of the gas sensor array by 25% and achieved a similarly high-performance classification and quantification to 16 unknown test samples, not used during the training of the CNN.

### 3.3. Experiment 3: Scenario 2 (Two Physical and Two Zero-Padded Sensor Elements)

In experiment 3, two of the physical sensor elements from the gas sensor array are replaced by zero-padded virtual sensor values. In totality, we still account for four-element data vectors after using zero-padding. Subsequently, spatially augmented data vectors achieved in this experiment are used for training and testing purposes accordingly. In this experiment, for all six possible cases, we achieved 100 percent classification accuracy while the obtained MSEs for quantification have been shown in Table 1.

The best performance was observed in the 6th case of scenario (ii) experiment. For quantification, MSE occurs 1.12 × 10^−2^, whereas sample-wise performances are shown in Figure 5. In this case, the maximum and minimum errors achieved 5.34 × 10^−2^ and 5.15 × 10^−6^. Therefore, it can be observed that after reducing two redundant sensor elements, it is possible to achieve 100 percent accurate classification. Better quantification performances can also be achieved without a significant decrease.

With two physical sensor element removal, we have reduced the power consumption of the gas sensor array by 50%, achieving similarly high-performance classification and quantification of 16 unknown test samples, not used during the training of the CNN.

With the results as discussed above, we achieve 100 percent classification accuracy in all eleven cases (baseline case, four cases in scenario 1, and six cases in scenario 2). Thus, the corresponding sample-wise classification performance is shown in Figure 6. It represents zero misclassification leading to 100 percent classification accuracy. Moreover, the MSEs for quantification in all the three experiments baseline, scenario 1, and scenario 2 for best-observed cases are shown in Figure 7.

The insight of redundancy is only drawn from the performance analysis, while Figure 2 explicitly shows the element-wise sensor characteristics only. Hence, from Table 1, MoO was found to be least affecting the performance and was considered the most redundant. Likewise, we found CdS as the next redundant sensor element to remove. The considered “baseline” is the reference performance of the unoptimized gas sensor system. In this work, we have proposed a method to reduce sensor node power by removing one or more gas sensor elements based on their information redundancy. Accordingly, in scenario 1, we have removed one gas sensor element out of the four. We have, hence, checked the performance of the CNN for every single element removal one by one. Similarly, in scenario 2, we have removed two gas sensor elements out of the four for all possible combinations and tested the resulting CNN for any performance degradation.

## 4. Discussion

As shown in the results, three experiments were performed. Experiment-1 demonstrated baseline results obtained using the actual data captured from four physical gas sensors consuming 10 Watts of effective power. With a four-element physical gas sensor array, we can classify all the test samples with 100% accuracy, while the quantification accuracy for each of the test samples has been achieved with squared error up to two to five decimal digits. With our proposed approach and use of CNN, the best performing sensor system now consumes only 5 Watts, achieving a power saving of 50%. Additionally, with the performed analytics, it is observed that sensor MoO is the most redundant, while sensor SnO2 is the most significant. Hence, sensor element MoO can be safely removed without significant performance degradation. Based on the results, there is minimal performance degradation when CdS and MoO are replaced with zero-padded virtual sensor response. Interestingly, the classification performance in all the test cases has been 100% accurate. It is recommended that with our approach, gas sensor nodes can be optimized to be power-efficient and suitable in a resource-constrained environment by harnessing the information redundancy present in a gas sensor array system.

6G-IoT has specifically considered the sensor nodes to be ultra-low power. It is one of the significant key performance indicators in the 6G-IoT paradigm. Accordingly, we have attempted to reduce the power requirement of a gas sensor array. This primary component consumes a more significant chunk of the power supply to the gas sensor node. A broad distribution of the power required in a gas sensor node is given in Figure 8 as a pie chart. The optimality of our proposed approach has been proven with the combination of power efficiency and high-performance results. The classification performance of an intelligent gas sensor system is measured in terms of its success in identifying respective test samples for the considered gases. This work has considered the performance achieved by using a four-element gas sensor array and suitable CNN as the baseline performance, achieving 100 percent correct classification of the considered test samples. This four-sensor element consumes 10 Watts of power. With the synergy of zero-padded virtual sensors, spatial augmentation, and CNN, we can still achieve 100 percent correct classification, even after removing two gas sensor elements from the four-element gas sensor array. With our proposed methodology, our proposed two-element gas sensor array consumes 5 Watts of power only. Therefore, we have identified removing the most redundant element, which does not degrade the performance, regarding removing any two sensor elements from the considered four gas sensor elements.

Our proposed scheme is a novel approach, and our predecessors have limited their works only up to high-performance classification using the considered dataset. Therefore, hardware reduction, cost reduction, consumed-power reduction, and high-performance are the merits of our proposed scheme in contrast with the existing schemes for the same purpose.

## 5. Conclusions

In the resource-constrained scenario in 6G-IoT systems, optimization of the gas sensor nodes can be achieved by removing redundant physical gas sensor elements. With the introduction of CNNs, any performance degradation due to the removal of physical sensor elements can be successfully mitigated in resource and data-constrained situations. Additionally, while removing redundant sensor elements, less hardware requirement without compromising the performance supports the sensor node applications orienting towards portable and wearable sensor technologies. The simpler 2D-CNN used in our experiment is computationally lightweight, making it more suitable in resource-constrained 6G-IoT scenarios. To the best of the author’s knowledge, this work is the first application of 2D-CNN to classify and quantify gases using zero padding and spatial augmentation. In data constrained situations, CNNs cannot otherwise be implemented due to the limited size of the data. Using our approach, with less hardware and power consumption, zero-padded virtual sensors with spatial augmentation leads to the CNN implementation to achieve higher performance. The gas sensor system, developed using our approach, can be conveniently designed in laboratory conditions and used in real-time applications supporting 6G-IoT in resource-constrained industrial and consumer sectors delivering high performance at the edge.

## Figures and Tables

**Figure 1 sensors-22-03039-f001:**
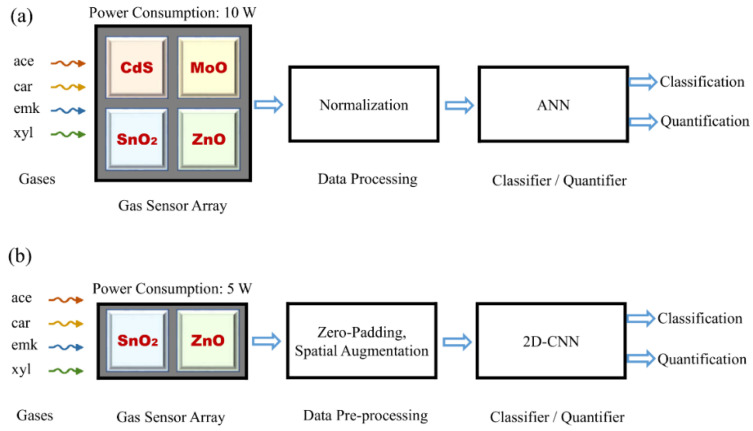
Schematic block diagrams of an electronic nose; using traditional ANN which are too complex due to usage of only fully connected layers resulting in a large number of trainable parameters (**a**); using the proposed sensor array optimization approach suitable for resource-constrained 6G-IoT scenarios incorporating a simpler 2D-CNN which results in less trainable parameters due to usage of convolutional layers than traditional ANN (**b**).

**Figure 2 sensors-22-03039-f002:**
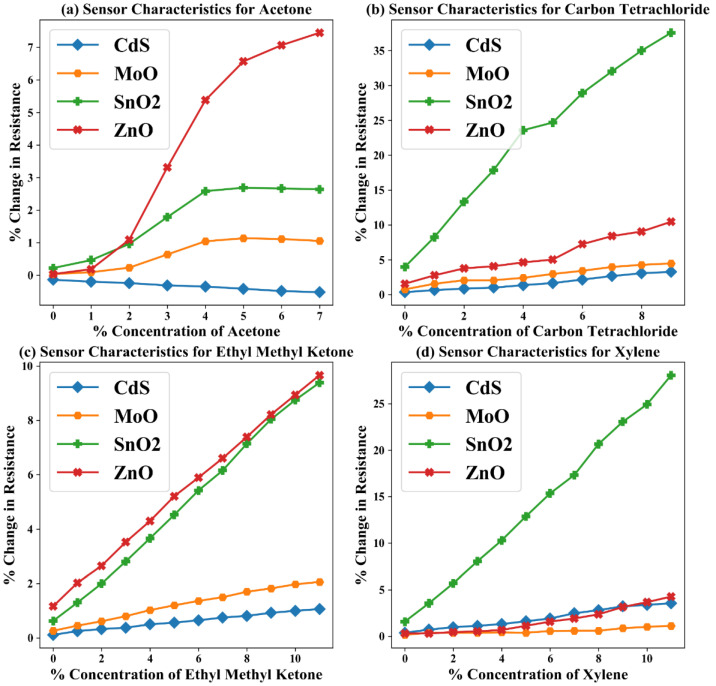
Percent change in resistance of sensor element vs. percent concentration of considered gas characteristics of four elements gas sensor array for four hazardous gases, viz., acetone, carbon tetrachloride, ethyl methyl ketone, and xylene [27].

**Figure 3 sensors-22-03039-f003:**
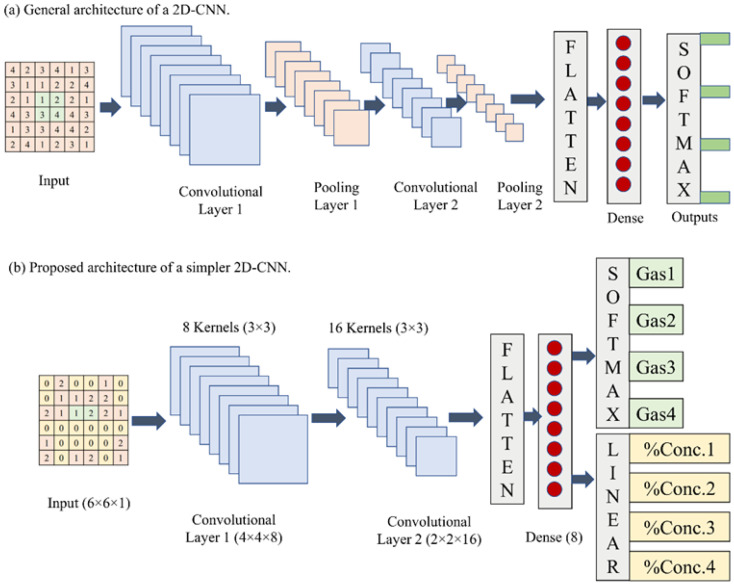
The general architecture of CNN consisting convolutional and pooling layers (**a**); Customization of the general architecture of a 2D-CNN for gas sensor node optimization suitable for resource-constrained 6G-IoT scenarios (**b**).

**Figure 4 sensors-22-03039-f004:**
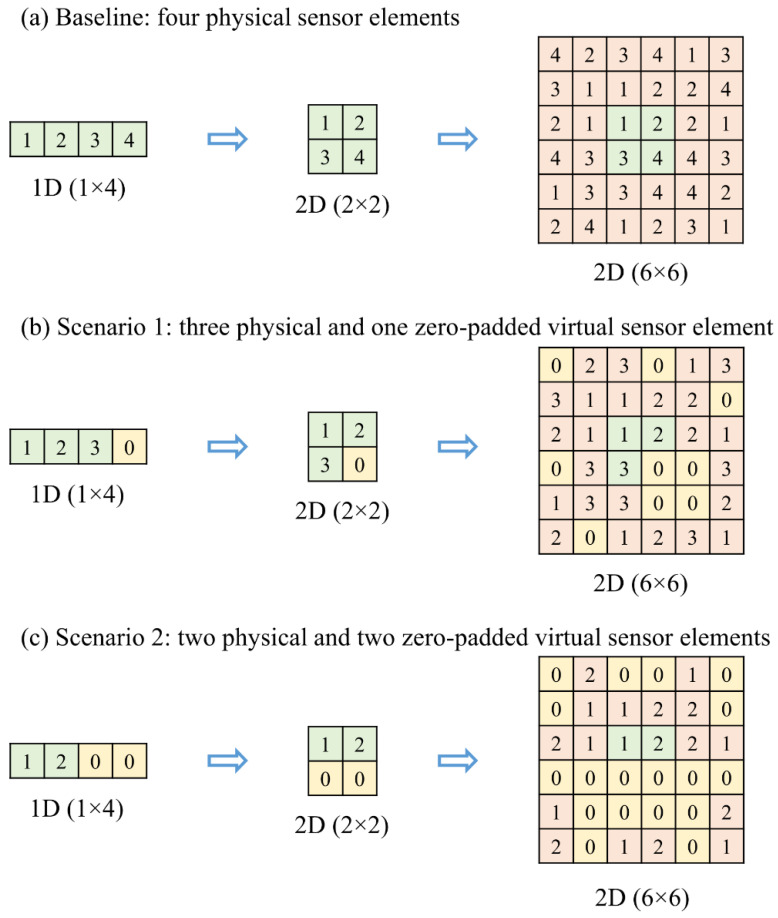
Sensor data vector augmentation procedure for; unoptimized gas sensor array (**a**); Scenario 1 when one redundant sensor element has been removed (**b**); Scenario 2 when two redundant sensor elements have been removed (**c**).

**Figure 5 sensors-22-03039-f005:**
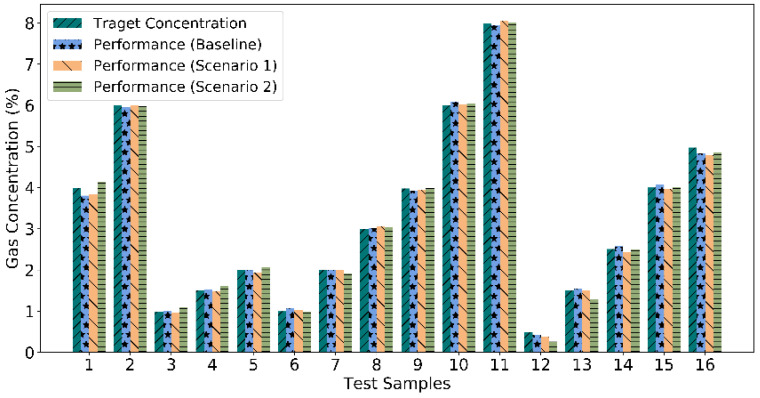
Quantification performance in terms of actual vs. predicted percent concentration of the considered gases for baseline, scenario 1, and scenario 2 experiments.

**Figure 6 sensors-22-03039-f006:**
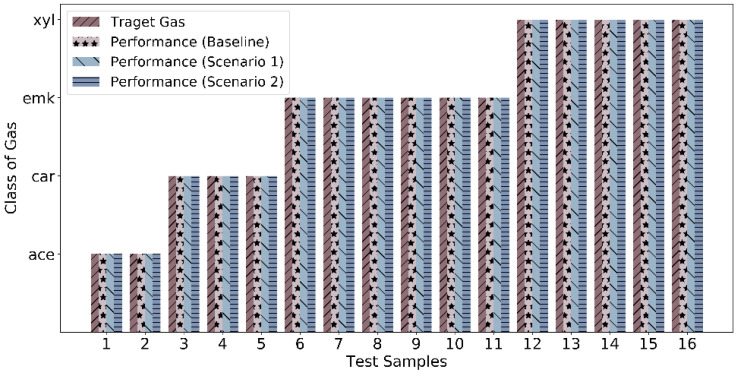
Classification performance in terms of actual vs. predicted target gas for baseline, scenario 1, and scenario 2 experiments.

**Figure 7 sensors-22-03039-f007:**
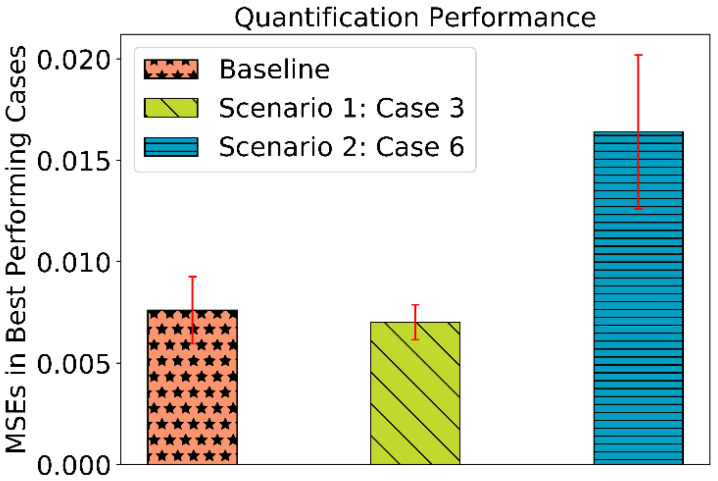
MSEs in quantification performance obtained in baseline and best performing cases in both the scenarios, viz., scenario 1: case 3 and scenario 2: case 6.

**Figure 8 sensors-22-03039-f008:**
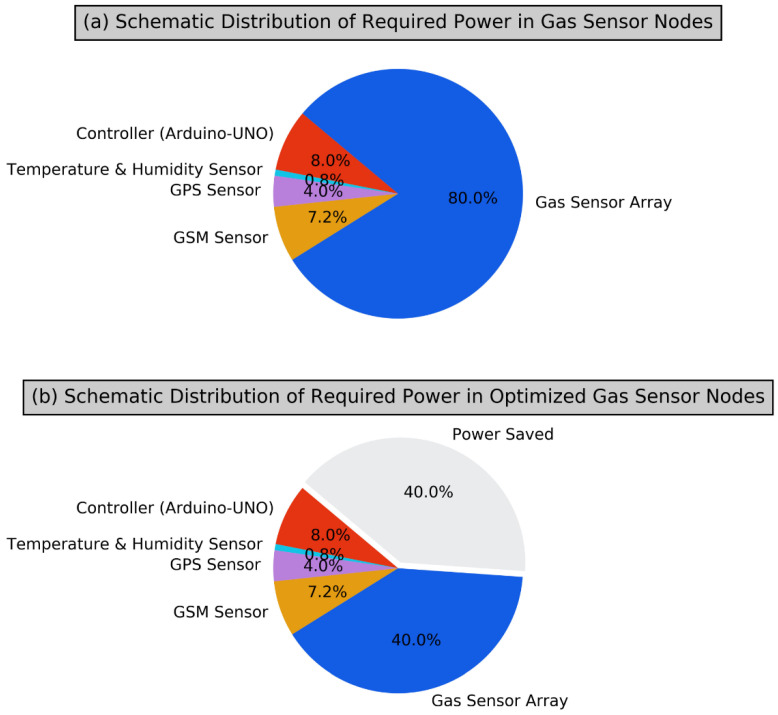
Schematic distribution of required power in gas sensor node; without optimization (**a**); with optimization carried out by our proposed approach (**b**).

**Table 1 sensors-22-03039-t001:** The obtained performance metrics in baseline, scenario 1, and scenario 2 experiments.

Experiments	Physical Sensors	Zero-Padded Virtual Sensors	Classification Accuracy (%)	Quantification MSE	Effective Power (Consumed)
Baseline	CdS, MoO, SnO_2_, ZnO	0	100	(7.61 ± 1.66) × 10^−3^	10 W
Scenario 1: Case 1	CdS, MoO, SnO_2_	1	100	(1.15 ± 0.35) × 10^−2^	7.5 W
Scenario 1: Case 2	CdS, MoO, ZnO	1	100	(4.01 ± 0.53) × 10^−2^	7.5 W
Scenario 1: Case 3	CdS, SnO_2_, ZnO	1	100	(7.02 ± 0.86) × 10^−3^	7.5 W
Scenario 1: Case 4	MoO, SnO_2_, ZnO	1	100	(1.43 ± 0.21) × 10^−2^	7.5 W
Scenario 2: Case 1	CdS, MoO	2	100	(3.01 ± 0.16) × 10^−0^	5 W
Scenario 2: Case 2	CdS, SnO_2_	2	100	(1.42 ± 0.33) × 10^−1^	5 W
Scenario 2: Case 3	CdS, ZnO	2	100	(2.43 ± 0.40) × 10^−2^	5 W
Scenario 2: Case 4	MoO, SnO_2_	2	100	(1.68 ± 0.19) × 10^−1^	5 W
Scenario 2: Case 5	MoO, ZnO	2	100	(3.17 ± 0.91) × 10^−1^	5 W
Scenario 2: Case 6	SnO_2_, ZnO	2	100	(1.64 ± 0.38) × 10^−2^	5 W

## Data Availability

The dataset used in this study is available permitted on request.

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
