# Peer review of "Zero-Padding and Spatial Augmentation-Based Gas Sensor Node Optimization Approach in Resource-Constrained 6G-IoT Paradigm"

_sensors, 2022, doi:10.3390/s22083039_

Round 1

Reviewer 1 Report

In this manuscript, the author proposed to use virtual gas sensors instead of physical gas sensors, and introduced Convolutional Neural Networks to classify and quantify gases, so as to reduce the total energy consumption of physical gas sensors in practical applications. However, the following issues limit the contribution of the manuscript.

The abstract is not well organized and should be further improved. Specifically, the authors should illustrate the contributions in abstract clearly.

The author claims that the proposed scheme can optimally reduce the power consumption of a gas sensor node without significant deviation in the node’s performance, and the so-called optimality is not shown by simulations. Moreover, it will be more convincing if the author can offer some theoretical proof to clarify the optimality of the proposed scheme.

The information and conclusions in Fig.1 and Fig.2 lacks necessary explanations, and influence of the parameters of the CNN model on the performance of the proposed should be illuminated.

Comparative analysis is missing. The authors should compare the proposed scheme with existing schemes.

In page 5, paragraph 2, according to the sensor characteristics given in Fig. 2, it can be known that CdS has higher redundancy for the three gases. However, the mean square error of MoO removal in Table 1 is the smallest. Please provide detailed analysis on this.

In page 6, paragraph 1, the authors are encouraged to introduce the parameters in the convolution process in details, such as step size.

On page 11, line 376, “sensor element MoO can be safely removed with significant performance degradation” should be “sensor element MoO can be safely removed without significant performance degradation”.

The authors should generally re-check for grammatical errors.

Reviewer 2 Report

This work investigates the application of 2D-CNN to classify and quantify gases using zero padding and spatial augmentation. The proposed approach of zero-padded virtual sensors with spatial augmentation leads to the CNN implementation to achieve higher performance with less hardware and power consumption. The topic of this paper is interesting; however, there are some concerns that need to be solved:

  1. This paper considers the 6G-IoT background. What’s the most unique feature of 6G-IoT, compared with traditional IoT? Moreover, the authors should explain why they consider the 6G-IoT scenario.
  2. The proposed hypothesis for gas sensor node power optimization has been tested under two scenarios. The authors should explain why they adopt these scenarios.
  3. The contributions of this paper is not well presented.
  4. 1 and 2 are not described, and the details of the figures are omitted.
  5. The baseline is too simple. I wonder whether there are some algorithms proposed by other researches that can be used as the baseline. Moreover, the presentation qualities of Figs. 5, 6, 7 are not good, since the figures look blurry.
  6. The authors should carefully proof the paper, correct the grammatical mistakes, and improve the writing quality.

Round 2

Reviewer 1 Report

The presentation of the paper should be improved.

Reviewer 2 Report

The presentation quality of this papaer needs to be improved.